# Genome-Wide Identification, Characterization, and Expression Analysis of GRAS Gene Family in Ginger (*Zingiber officinale* Roscoe)

**DOI:** 10.3390/genes14010096

**Published:** 2022-12-29

**Authors:** Shuming Tian, Yuepeng Wan, Dongzhu Jiang, Min Gong, Junyao Lin, Maoqin Xia, Cuiping Shi, Haitao Xing, Hong-Lei Li

**Affiliations:** 1College of Landscape Architecture and life Science/Institute of special Plants, Chongqing University of Arts and Sciences, Chongqing 402168, China; 2College of Biology and Food Engineering, Chongqing Three Gorges University, Chongqing 404020, China; 3Chongqing Key Laboratory of Economic Plant Biotechnology, Chongqing University of Arts and Sciences, Chongqing 402160, China

**Keywords:** ginger, GRAS family, gene expression, phylogenetic analyses

## Abstract

GRAS family proteins are one of the most abundant transcription factors in plants; they play crucial roles in plant development, metabolism, and biotic- and abiotic-stress responses. The GRAS family has been identified and functionally characterized in some plant species. However, this family in ginger (*Zingiber officinale* Roscoe), a medicinal crop and non-prescription drug, remains unknown to date. In the present study, 66 GRAS genes were identified by searching the complete genome sequence of ginger. The GRAS family is divided into nine subfamilies based on the phylogenetic analyses. The GRAS genes are distributed unevenly across 11 chromosomes. By analyzing the gene structure and motif distribution of GRAS members in ginger, we found that the GRAS genes have more than one *cis*-acting element. Chromosomal location and duplication analysis indicated that whole-genome duplication, tandem duplication, and segmental duplication may be responsible for the expansion of the GRAS family in ginger. The expression levels of GRAS family genes are different in ginger roots and stems, indicating that these genes may have an impact on ginger development. In addition, the GRAS genes in ginger showed extensive expression patterns under different abiotic stresses, suggesting that they may play important roles in the stress response. Our study provides a comprehensive analysis of GRAS members in ginger for the first time, which will help to better explore the function of GRAS genes in the regulation of tissue development and response to stress in ginger.

## 1. Introduction

Transcription factors (TFs) are extraordinary proteins that participate in regulating plant growth, development, signal transduction, and resistance to biotic and abiotic and stresses [1]. TFs modulate the expression of target genes by combining with their *cis*-acting regulatory elements [2,3]. The identification and analysis of TFs are fundamental in functional genomic research. Since the first transcription factor was discovered in maize [4], more than 3000 TFs have been demonstrated to be involved in various physiological processes and regulatory networks in plants [2]. Among them, some important transcription factors, including AP2/EREBP, WRKY, bZIP, MADS, MBS, ARF, HB, SBP, and others, have been well studied [5,6,7,8].

The GRAS gene family is an important plant-specific transcription factor that is named after the three earliest identified members: GAI (gibberellic acid insensitive), RGA (repressor of GAI), and SCR (scarecrow) [9,10]. These proteins were first found in bacteria and were transferred into plants via the mechanism of lateral gene transfer [11]. In general, GRAS proteins contain 400 to 700 amino acid (aa) residues, which are mainly located in the nucleus [12]. The C-terminal region of GRAS genes is very conserved and forms the GRAS domain. The GRAS domain has five motifs: VHIID motif, PFYRE motif, Leucine heptad repeat II (LHR II), Leucine heptad repeat I (LHR I), and the SAW motif [13]. According to related studies, the GRAS gene family has been identified in a variety of plants so far, including poplar, Arabidopsis, rice, plum, and pine [14,15,16,17]. There are 35 and 61 members of the GRAS gene family in the model plant Arabidopsis and rice genomes, respectively. These GRAS proteins can be divided into at least eight subfamilies: SCR, LAS, DELLA, PAT1, SHR, HAM, SCL3, and LISCL [18,19]. Based on the accumulation of data on the GRAS family, more than 13 clades were identified in plants [20,21,22]. The evolutionary history of the GRAS family in plants has been studied extensively so far, including the model species, Arabidopsis [19] and tobacco [23], as well as the common crops rice [23], maize [24], sweet potato [25], cotton [5], sativa [19], castor [26], poplar [27], cabbage [24], and grape [28].

The members of GRAS family play important roles in diverse processes such as plant growth and development, and tolerance to biotic or abiotic stresses [29,30,31,32,33,34]. Each subfamily of GRAS has unique functions, whereas the proteins from the same subfamily have similar functions. Genes of the DELLA subfamily have been shown to be blockers of the gibberellin (GA) signaling pathway, e.g., GAI, RGA, and RGL [32]. GAI and RGA act as negative regulators of GA signaling and control various plant developmental processes, including seed development and vegetative reproductive-phase transition in Arabidopsis [35]. It has been reported that members of the SCR and SHR subfamilies can form the SCR/SHR complex, which is involved in regulating the radial growth of root [36]. The subfamily LAS genes are always in the hub of the gene regulatory network and can downregulate GA content [37]. When *At*LAS was knocked out in Arabidopsis, lateral roots could not be formed [31]. The apical meristem organization and growth of *Capsicum annuum* were regulated by members of the subfamily HAM. The PAT1 subfamily plays important roles in activating the ROS scavenging system, which could improve tolerance to salinity, drought, and cold [38,39]. SCL proteins display unexpected tolerance to diverse environmental stresses; for example, SCL14 is involved in promoting the resistance of Arabidopsis through activating stress-inducible promoters [40]. The drought and salt resistance were enhanced by the overexpression of the *PeSCL7* gene in Arabidopsis [41]. Moreover, when the *ZmSCL7* gene was overexpressed in tobacco, plant resistance to salt was significantly improved.

Ginger (*Zingiber officinale* Roscoe) is an herbaceous perennial plant that belongs to the family Zingiberaceae. Ginger is one of the most widely cultivated crops in numerous countries, including China, India, and Nigeria. The global ginger production in 2019 was up to 4.08 million tons according to data from the FAO. The rhizome of ginger is a popular spice and traditional Eastern medicine [42,43]. The bioactive components of ginger have shown therapeutic properties as an agent for treating rheumatoid arthritis, asthma, and other ailments [44,45]. A number of TFs have been reported to be crucial in regulating the rhizome development and abiotic stress. For example, AUX/IAA and MADS box protein respond to rhizome initiation and development [46]. The ZoAP2/ERF family genes play a crucial role in tolerance to cold, drought, and salt [47]. However, the function of the GRAS family in ginger remains unknown. Based on the published whole genome of ginger [48], this study identified and analyzed the ginger GRAS family genes by using bioinformatic methods. The present systematic research provided some insights that could be used in elucidating the function of ginger GRAS.

## 2. Materials and Methods

### 2.1. Genome-Wide Identification of GRAS Genes in Ginger

The full genome sequence and annotation data of *Arabidopsis thaliana* and rice were obtained from the Phytozome 13. The ginger genome was sequenced by our group and was published in *Horticulture Research* [48]. Moreover, all the Arabidopsis and rice GRAS genes were obtained from Plant TFDB, based on previous reports. In order to identify out all the possible GRAS genes in ginger, the BLASTP search against all the protein sequences of ginger was performed using all the GRAS sequences of both Arabidopsis and rice as inquires. GRAS sequences of ginger were extended by combining gene annotation information and BLASTP search. To confirm the accuracy of predicted ginger proteins, we used the NCBI website to evaluate whether the candidates contain no GRAS structural domains or less than half the length of a typical GRAS structural domain (350 aa) [49]. Those candidates lacking a typical conserved GRAS domain were treated as non-redundant GRAS member and excluded. The ExPASy software (https://web.expasy.org/protparam/, accessed on 16 December 2022) was used to analyze the composition, chemical, and physical characterization of ginger GRAS proteins [50]. WoLF PSORT (http://wolfpsort.org, accessed on 16 December 2022) was used to predicate the subcellular localization of the identified GRAS proteins.

### 2.2. Phylogenetic Analysis of the GRAS Genes

The phylogenetic analysis was conducted based on all of the GRAS protein sequences of ginger, *A. thaliana* and rice. The Clustal X software was used to align the GRAS sequences through the default parameters. To investigate the relationships of GRAS members, the phylogenetic analysis was performed using the maximum likelihood (ML) method in MEGA 11 (https://www.megasoftware.net/, accessed on 21 May 2022), with 1000 bootstrap replications [51]. Subsequently, these GRAS sequences were classified into different subfamilies on the basis of the obtained phylogenetic tree. At last, the phylogenetic tree was edited and visualized.

### 2.3. Chromosomal Location and Gene Duplication

The chromosome positions of 66 GRAS genes were obtained from ginger transcriptome of the research group. Duplicate gene pairs were generated by BLAST program through aligning the paralogs of ginger GRAS. The multiple collinear scanning tool kits (MCScanX) was used to analyze the gene duplications and collinearity relationships of GRAS genes [52]. If both coverage and similarity of aligned genes are greater than 75%, potential gene duplication events will be hypothesized to occurred among these genes. Tandem duplications were defined as duplicated genes with less than one intervening on a single chromosome. If not, they were treated as segmental repeats in the present study. We used TBtools software to draw the physical map of GRAS genes and display the distribution of GRAS genes on different ginger chromosomes.

### 2.4. Gene Structure and Conserved Motif Analysis of GRAS Genes in Ginger

In order to determine the features in the ginger GRAS sequences, multiple sequence alignment of 66 GRAS proteins was obtained by using Jalview software v2.10.5 with default parameters and realigned using Clustal. The conserved motif domains of ginger GRAS proteins were examined by using the MEME suite tool (http://meme.nbcr.-net/meme/intro.html, accessed on 23 May 2022) from the website [53]. The exon–intron structure of ginger GRAS was analyzed by using the TBtools based on the gff3 file of ginger genome. The patterns of gene structure and conversed motif were visualized by TBtools.

### 2.5. Expression Analysis of Ginger GRAS Genes 

The transcriptome data was used to reveal the expression patterns of GRAS genes in different tissues. Young leaves of ginger after salt, heat and cold treatments were collected for RNA-seq. Total RNA was extracted by using the TRIzol kit (Invitrogen, Carlsbad, CA, USA) an mRNA purification kit (Promega, Beijing, China) was used to purify mRNA from total RNA, according to the manufacturers’ protocols. For each sample, an amount of 20 μg RNA was enriched by using oligo (dT) magnetic beads. Then, these RNAs were digested into short fragments. The first- and second-strand cDNA were synthesized and purified at the BGI (Shenzhen, China). The purified fragments were linked to the sequencing adaptors. The RNA-seq was carried out on an Illumina Hiseq 2000 sequencing system. Differentially expressed genes were identified by a rigorous algorithm. The significance of gene expression difference was judged by a criteria of the absolute value of log2 ratio > 1 and *p* < 0.001 based on the *Z. officinale* genome sequence.

QRT-PCR was used to detect the expression of selected GRAS genes in response to abiotic-stress. The qRT-PCR primers were designed by using Primer Primer 5 software (http://frodo.wi.mit.edu/, accessed on 23 May 2022) (Appendix A Appendix A). The TUB*2* gene is always used as an internal reference gene; it was found to be expressed in most of the tissues with stable expression levels. The TUB*2* gene was used as an internal control in this study. The PCR program was was performed as follows: an initial denaturation of 95 °C for 30 s, a second denaturation of 95 °C for 10 s 40 cycles, and 60 °C for 30 s. Each reaction was carried out with three biological replicates. The 2^−△△CT^ method was used to calculated the relative expression level of each GRAS gene [54].

### 2.6. cis-Elements and Target miRNAs Analysis of Ginger GRAS

For each *ZoGRAS* gene, 2 kb (kilo-base) DNA sequences in upstream regions were extracted from the ginger genome. The *cis*-regulatory elements were searched and analyzed in this 2 kb region by using the online server of PlantCARE (http://bioinformatics.psb.ugent.be/webtools/plantcare/html/, accessed on 23 May 2022) and TBtools v1.0971 [52]. The web-based psRNA Target Server (http://zhaolab.org/psRNATarget, accessed on 27 December 2022) was used to determine the potential miRNAs that targeting the *ZoGRAS* genes with default parameters.

## 3. Results

### 3.1. Identification and Physicochemical Properties of GRAS Genes in Ginger

In total, 66 GRAS proteins in ginger were identified by a combination of methods (Appendix A Appendix A). The complete GRAS domain (PF03514) were found in all of the ginger GRAS proteins. The physical and chemical characteristics of ginger GRAS proteins were analyzed by using ExPasy. Among the 66 GRAS proteins, *Zo*GRAS#34 is the smallest, with 381 aa; the largest is ZoGRAS#13, with 3159 aa. The molecular weight of the ginger GRAS proteins ranged from 14.66 kDa (ZoGRAS#34) to 117.37 kDa (ZoGRAS#13). The pI ranged from 4.81 (ZoGRAS#34) to 9.65 (ZoGRAS#37). The numerical range of molecular weight and PI agree with that of other species, indicating the conservation of GRASs in different species. Subcellular localization analysis showed that a total of 30 GRAS proteins were located in the nuclear region, 16 in the cytoplasm, 1 in the intercellular filaments, and 19 in the chloroplasts (Appendix A Appendix A).

### 3.2. Multiple Sequence Alignment, Phylogeny, and Classification of GRAS Genes

In order to detect the relationships among GRAS family proteins, the GRAS domains of 70 rice, 48 Arabidopsis and 66 ginger GRASs were collected. The aligned GRAS data were used to construct the phylogeny using the Maximum Likelihood method in MEGA (Figure 1 and Appendix A Appendix A). According to the relationship with known rice and Arabidopsis homologs, and the clustering patterns here, ginger GRAS proteins were classified into nine subfamilies: SCL3, DELLA, HAM, LISCL, SHR, PAT1, LAS, SCR, and DLT. SHR and PAT1 were the most abundant subfamilies with 12 and 13 GRAS genes, respectively. The other subfamilies, namely SCL, DELLA, HAM, LISCL, LAS, SCR, and DLT, had 4, 6, 11, 7, 2, 9, and 2 GRAS genes, respectively.

### 3.3. Chromosome Distribution of Ginger GRAS Genes

According to the physical location of genes in the ginger genome, the chromosomal positions of the GRAS gene were depicted (Figure 2). Our results showed that the 66 GRAS genes distributed unevenly on the 11 ginger chromosomes. Most of the *ZoGRAS* located in the areas with high gene density. It is worth noting that Chr04 contained the most GRAS genes (9 genes). Both Chr03 and Chr01 have 8 GRAS gene members. Some chromosomes (e.g., Chr06, Chr07 and Chr06) have less than 5 GRAS genes.

### 3.4. Gene Structure and Motif Analysis of Ginger GRAS

Exon/intron gene structure plays an important role in gene family evolution. GRAS intron and exon structure were obtained by comparing the genomic DNA sequences (Figure 3). The positioin and number of exons and introns of all GRAS genes were analyzed. Our analysis showed that 77.3% of the GRAS genes had no intron. Nine *Zo*GRAS members contain one or more introns. GRAS13 contains 15 introns. Three introns were found in *Zo*GRAS49, *Zo*GRAS55, and *Zo*GRAS15.

There are 10 conserved motifs (Motif 1–10) in ginger GRAS proteins. The number of motifs vary among ginger GRAS proteins. The conserved motifs of all GRAS proteins were mainly found in the C-terminus, and the types and positions of conserved motifs contained in the same subfamily of proteins were very similar (Figure 4). For example, the DELLA subfamily proteins all contain conserved motifs, and the SCR subfamily proteins all contain conserved motifs. In ginger, Motif-1, Motif-2, and Motif-7 were identified in most of the GRAS proteins. ZoGRAS#34 contains 3 conserved motifs: Motif 3, Motif 5, and Motif 8. ZoGRAS#39 and ZoGRAS#38 contain 4 motifs: Motif 1, Motif 2, Motif 7, and Motif 9. ZoGRAS#60 contains 5 motifs: Motif 2, Motif 3, Motif 6, Motif 8, and Motif 9. ZoGRAS#36 contains 5 motifs: Motif 1, Motif 2, Motif 7, Motif 8, and Motif 9. The number of conserved motifs on the rest of GRAS proteins in ginger ranged from 6 to 10. Based on the analysis of the structure and conserved motifs of GRAS proteins in different subclades, it is clear that members within the same subclade are relatively conserved. Thus, the functions of genes in the same subclade may be similar.

### 3.5. GRAS Promoter cis-Acting Elements and Target miRNA Analysis

In order to further study the regulation mechanism of ginger GRAS genes under abiotic stress, the upstream 2 kb sequences of 66 GRAS genes were captured from ginger genome to analyze the cis-acting elements. The results show that there are approximately 30 cis-acting elements that can function efficiently, and cis-acting elements that can be efficiently expressed, as well as elements with well-defined functions, were analyzed and explained. Hormone response elements, including ABRE (abscisic acid responsive), P-box, GARE-motif and TATC-box (gibberellin responsive), TCA-element and SARE (salicylic acid responsive), TGACG-motif and CGTCA-motif (MeJA-responsive), TGA-box (auxin-responsive), and Unnamed_1 (phytochrome) (Figure 5), were identified. Abiotic-stress-response elements include TC-rich repeats (defense and stress responses), GC-motif (hypoxia-specific induction), MBS (drought inducibility), WUN-motif (wound responsive element), DRE, and LTR (dehydration, low temperature, and salt-stress responsiveness). Response elements to light include the TCT-motif, ACE, GT1-motif, G-Box, Sp1, 3-AF1 binding site, AAAC-motif, 4cl-CMA1b, MRE, Box 4, ATCT-motif, ATC-motif, CAG-motif, TCCC-motif, LAMP-element, GATA-motif, ACE, I-box, AE-box, ACA-motif, AT1-motif, BoxII, chs-CMA1a, chs-Unit 1 m1, C-box, chs-CMA2a, Gap-motif, GA-motif, L-box, Pc-CMA2c, sbp-CMA1c, and TCT-motif. Other response elements include the CCAAT-box, Unnamed__1, AT-rich element, Box-III, HD-Zip 3 (protein binding site), NON-box (meristem expression) and CAT-box, TATA-box, motif I (root specificity), CAAT-box and A-box (promoter and enhancer), AT-rich sequence (ecitator), MBST (flavonoid biosynthesis gene regulation), MSA-like (cell cycle), GCN4_motif (endosperm expression), ARE (anaerobic induction), O2-site (zein metabolic regulation), HD-Zip 1 (differentiation of palisade mesophyll cells), circadian (circadian rhythm adjust) and RY-element (seed-specific regulation). A WUN-motif associated with wound reactivity was found in both *Zo*GRAS#19 and *Zo*GRAS#11. The RY element, a seed-specific element, is found only in *Zo*GRAS#49 and *Zo*GRAS#3. Motif I is a root-specific regulated cis-element found in *Zo*GRAS#53. MBST is a flavonoid biosynthesis gene-regulated cis-element found in *Zo*GRAS#42, *Zo*GRAS#58, *Zo*GRAS#26, and *Zo*GRAS#20. MSA, which is associated with the cell cycle, is found in *Zo*GRAS#25, *Zo*GRAS#48, *Zo*GRAS#47, *Zo*GRAS#46, *Zo*GRAS#66, and *Zo*GRAS#65. Presumptive target gene prediction analyses showed that 19 target miRNAs for 11 *Zo*GRAS genes were predicted (Appendix A: Appendix A).

### 3.6. GO Annotation of GRAS Protein Sequences

To understand the functions of GRAS proteins in different biological processes in ginger, a GO annotation analysis of ZoGRAS genes were performed. The results suggest that GRAS proteins may be involved in many biological, cellular, and molecular processes. Most GRAS proteins have the function of protein-binding and transcriptional regulation. The major GRAS proteins are involved in the regulation of RNA biosynthesis processes, nucleic acid template transcription, and template transcription. There are some genes involved in hormonal regulation, such as gibberellin regulation. Some genes were also found to be involved in regulating plant organs development and abiotic stresses response. The results showed that 50 of the 66 GRAS proteins have organic cyclic compound binding, specific binding, and transcriptional regulatory activities. The biological process analysis of GRAS genes showed that GRAS genes function in biometabolic, abiotic, and biotic-stress response (Figure 6).

### 3.7. Synteny and Evolutionary Analysis of GRAS Genes

In all organisms, gene replication happens frequently, causing functional changes of new genes from previous genes. In a previous study, tandem duplication event was defined as more than one gene family member arising within a 200 kb intergenic region. In contrast, segmental duplication is common in plants. Because the process of polyploidization make these plants retain many large blocks of duplicated chromosomes. In general, the copy number increase of gene family in plants was contributed by tandem and segmental duplications. Many ginger GRAS genes exist on different chromosomes in ginger, indicating that the GRAS gene family is highly conserved (Figure 7). In this study, two GRAS genes resulted from one tandem duplication event (*Zo*GRAS#18/*Zo*GRAS#2*0*) were found in ginger chromosome (Chr04). In total, 50 ginger GRAS genes associated with 25 segmental duplication events were detected, which indicate that segmental duplication might drive the evolution of ginger GRAS family. The segmental events were assessed to have occurred 4.66 million years ago (Ma).

To explore the relationships of the ginger GRAS family with other plants, two comparative collinear maps of ginger related to Arabidopsis and barley were constructed (Figure 8). Homology of ginger GRAS genes were found in both Arabidopsis and barley. In total, 6 GRAS genes of ginger displayed syntenic relationships with those in Arabidopsis. However, no syntenic gene pairs between barley and ginger were found.

### 3.8. Expression Patterns of Ginger GRAS Genes in Response to Abiotic Stress

In order to study the potential functions of the GRAS genes under various non-biological stresses, RNA-seq data under heat, cold, and salt treatments were used to detect their expression levels. In total, 66 GRAS genes were found differentially expressed in at least one stress treatment. The expression of 12 randomly selected genes were also detected by using qRT-PCR under cold- and heat-stress treatments. All the 12 selected genes are significantly induced by stress at one or more time points (Figure 9). In general, genetic response was slowe under cold conditions. The gene expression level increased gradually under low-temperature conditions and peaked at 12 h or 24 h (Figure 10).

### 3.9. GRAS Gene Expression Profile of Ginger in Different Tissues

In order to explore the potential functions of the ginger GRAS genes in different developmental stages of ginger organs, RNA-seq data were used to examine their expression patterns (Figure 11). A total of 39 of the 66 GRAS genes were found expressed in all samples (FPKM > 0), and 27 GRAS genes exhibited constitutive expression (FPKM > 1 in all samples). Some genes showed preferential expression in the tissues detected. For example, one gene in the meristem of stems (*ZoGRAS#45*), two genes in leaves (*Zo*GRAS#9), a couple of genes in mature florescences (*Zo*GRAS#9 and *Zo*GRAS#45), and three genes in the roots (*Zo*GRAS#44, *Zo*GRAS#*9*, and *Zo*GRAS#45) exhibited highest expression level. The expression level of some genes showed a obvious trend in different development stages of ginger organs. Such as, the expression level of *ZoGRAS#9* gradually decreased, whereas, the expression level of *ZoGRAS#19* showed a increasing trend (Figure 11). The transcriptional abundance of GRAS genes varied in different organs, suggesting that GRAS genes exhibit diverse functions during the growth and development of ginger.

## 4. Discussion

Transcription factors play important roles in plant growth and development, response to adversity stress, and various aspects of plant life activities. Previous studies have also shown that the expression of GRAS family members in different organs e.g., roots, stems, leaves, flowers, and fruits varies during plant growth and development, thus indicating that the GRAS gene family is involved in all stages of plant growth and development [29,30,31,32,33,34]. Moreover, the GRAS gene family plays a key regulatory role in response to environmental stresses [3,20,26,28]. Nevertheless, the GRAS genes in ginger have not been reported to date. In this study, we searched GRAS genes in the ginger genome and identified 66 GRAS family genes, which is more than the number of GRAS genes in Arabidopsis (48) and tomato (53), but less than the number of GRAS genes in rice (70) [55]. Previous studies have suggested that GRAS proteins may expand after the divergence of higher and lower plants, and their numbers vary widely in different plants [3,24]. Comparative analyses have shown that amplification of the GRAS gene family is associated with whole-genome duplications, tandem duplications, and segmental duplications [12]. Tandem repeats were detected in Arabidopsis, tomato, rice, and poplar [22,27]. In ginger, a whole-genome duplication event occurred during the evolutionary history [48]. In this study, chromosome mapping showed that *Zo*GRAS genes are distributed on all the 11 chromosomes of ginger, whereas the number of *Zo*GRAS ranges from 4 to 9. A total of 1 tandem duplication event and 25 segmental duplication events in connection with 52 GRAS genes were determined. Thus, expansion of the *Zo*GRAS family in ginger might be the result of whole-genome duplication, segmental duplication, and tandem duplications.

In this study, the GRAS genes of Arabidopsis, rice, and ginger were divided into nine clades. The relationship between ginger and *Arabidopsis thaliana* suggests the origin and diversity of GRAS. The results indicate that the *Zo*GRAS genes originated and differentiated in the ancestor of monocotyledons (ginger) and dicotyledons (*A. thaliana*). The PAT1 subfamily contains the largest number of ginger GRAS genes (15). The genes of the PAT1 subfamily play an important role in plant phytochrome and defense signaling pathways [10,56]. SHR and SCR subfamilies are involved in maintaining root apical meristem and regulating the root morphology in Arabidopsis [57]. Therefore, 19 *Zo*GRAS genes (12 homologous to *At*SHR and 7 homologous to *At*SCR) may have similar functions. The HAM subfamily, consisting of four *Zo*GRAS proteins, could be involved in shoot meristem formation [32]. In ginger, six *Zo*GRAS genes were identified as DELLA subfamily members. Studies have shown that the DELLA subfamily plays a negative regulatory role in the GA pathway and could lead to the dwarfing of plant growth [58]. The phylogenetic analysis of the GRAS gene family provides a theoretical basis for further study of the functional genome of ginger.

Introns play a very important role in plants’ evolution and undergo loss and gain during the evolutionary history [59]. The GRAS gene has been identified in 67% of individual exon-less genes in Arabidopsis [27], 84% in poplar [60], and 77.4% in tomato [49]. The proportion of intron-less GRAS genes in ginger was higher (~77.3%) than in Populus (54.7%), rice (55%), and Arabidopsis (67.6%), but lower than in Plum (82.2%) [24,61,62]. Intron-less genes have also been identified in other large gene families, e.g., the small auxin-up RNA (Saur) gene family and the DEAD box RNA helicase F-box TF family [63,64]. Plant GRAS genes were hypothesized to originate from prokaryotic genes through a mechanism of horizontal gene transfer and duplicate during evolution, which could explain the scenery of abundant GRAS genes without introns [27,65]. Intron-less genes may be able to respond quickly to stress and regulate growth and developmental processes in plants [66,67,68]. Thus, many *Zo*GRAS members tend to respond rapidly to environmental changes. We analyzed 10 different conserved motifs in the *Zo*GRAS family and found that members in the same group usually have a similar motif composition and transcriptional regulators, whereas the motif compositions and distributions in the N-terminus vary remarkably among different *Zo*GRAS subgroups. In evolutionary processes, mutations of non-conservative amino acids cause motifs’ variation, which contributes to the distinct and diverse gene functions of GRAS genes.

GRAS proteins exhibit multiple functions and play important roles in plant growth and a variety of physiological processes, including GA signal transduction and hormone metabolism regulation in response to biotic and abiotic stresses [13,69]. The DELLA protein plays a negative role in the endogenous plant hormone gibberellin, leading to the reduced sensitivity of plants to gibberellin, resulting in a range of phenomena such as dwarfism during growth [70]. In addition, the study of the PATl family revealed the response of the plant PATl family to light [10]. Both PAT1 and SCL13 were found to act in the light signaling pathway of Arabidopsis and control the basic plant hairs through photoreceptor transduction signals [56]. Tissue-specific expression patterns indicated that most of the ZoGRAS genes were expressed in all tested tissues. In the GO enrichment analysis, the genotype ends are subject to functional diversity, resulting in the majority of ginger GRAS proteins playing important roles in many different biological processes. Based on the analysis of the promoted region, we found growth-related, abiotic-responsive, and hormone-responsive cis-elements in the promoted region. These results suggest that ginger GRAS transcription factors play important roles in hormone signaling pathways. The comparison of GRAS homologs in different species, including expression patterns and protein sequences, has enhanced our understanding of the role of these GRAS genes in ginger.

Transcriptional regulators belonging to the consent subcluster share a recent common evolutionary origin and have molecularly functionally related conserved motifs. Highly homologous genes between ginger and rice and Arabidopsis were used to predict gene function in ginger. No studies have been conducted to analyze the expression pattern of ginger GRAS family genes under abiotic stress. In this study, we analyzed the expression patterns of ginger GRAS family genes in different tissues, different development stages, and different abiotic stress treatment with transcriptome and real-time qPCR. In Arabidopsis, a GRAS family transcription factor called SCARECROW-LIKE28 (SCL28) plays a critical role in determining cell size [71]. We found that *ZoGRAS#66* and *ZoGRAS#65* in ginger are closely related to *AtSCL28*, thus indicating that these two genes may be involved in regulating cell size, which is associated with the expansion of the ginger rhizome. *AtSCL15*, a member of the HAM subfamily, is involved in salt- and cold-stress responses [72]. In particular, three genes of the HAM subfamily in ginger, namely *ZoGRAS#13–15*, were also found to be significantly differently expressed in cold- and salt-stress treatment. Previous studies have shown that *AtSCL14* genes of the LISCL subfamily in *A. thaliana* can enhance stress resistance through upregulating the expression of stress-response genes [66]. Similarly, four genes (*ZoGRAS#23–25*,*27*) of the LISCL subfamily were significantly upregulated under cold stress. A member of the PAT1 subfamily *AtSCL13* showed strong tolerance to drought-, cold-, and salt-stress treatments [66,73]. In this study, six *ZoGRAS* genes of the PAT1 subfamily were found to be differently expressed. Among them, *ZoGRAS#42* and *ZoGRAS#48* were significantly upregulated under heat treatment. *ZoGRAS#48*, *ZoGRAS#52*, and *ZoGRAS#53* increased significantly after cold and salt treatments. In general, functions of *ZoGRAS* genes overlap in tissue development and response to various stresses; this finding is in agreement with previous studies, as reviewed by Waseem et al. [3]. These results lay a foundation for further study of stress response and additional functions of ZoGRAS gene family members.

## 5. Conclusions

In this study, a comprehensive analysis of the GRAS family genes in ginger was performed, and a total of 66 full-length GRAS genes were identified. Based on the phylogenetic tree and the distribution of conserved motifs, the GRAS family was divided into 10 subfamilies with highly similar gene structures and motif compositions in the same subfamily or subgroup. The phylogenetic comparison and collinear analysis of different plant GRAS genes provided important clues for the evolutionary characteristics of ginger GRAS genes. The GRAS gene plays an important role in the growth and development of ginger. The phylogeny and gene expression analysis and abiotic stress treatment during the development of ginger rhizome will provide a reference for the functional analysis of the GRAS gene. Taken together, these results provide a valuable resource for a better understanding of the biological role of the GRAS gene in ginger. In the future, various advanced approaches, e.g., yeast hybrids, co-immunoprecipitation, and yeast pull-down assay, are needed to uncover the mechanism of GRAS TFs in ginger development and stress.

## Figures and Tables

**Figure 1 genes-14-00096-f001:**
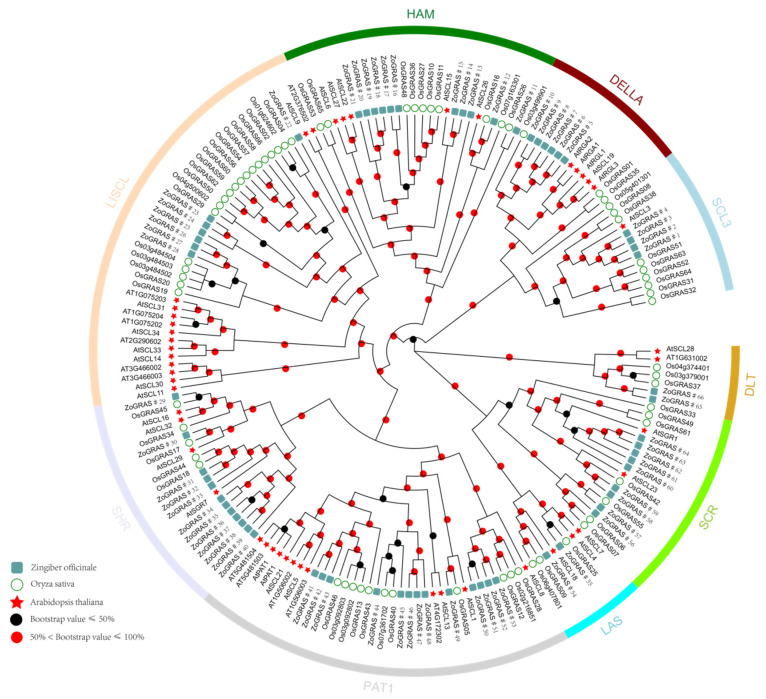
A rootless phylogenetic tree representing 66 GRAS proteins in ginger, rice, and Arabidopsis. The arcs in different color indicate different subfamilies of GRAS. Ginger GRAS proteins with the prefix “Zo” indicate “*Zingiber officinale*”.

**Figure 2 genes-14-00096-f002:**
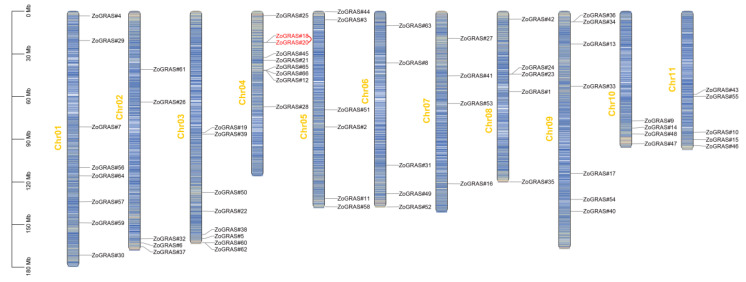
The chromosomal distribution patterns of ginger GRAS genes. GRAS gene pairs resulted from tandem duplication event are connected with red lines. Characters in yellow are the chromosome names. The color gradient of chromosomes from blue to red represent gene density from low to high. Blank area represents lack of gene distribution on chromosomes.

**Figure 3 genes-14-00096-f003:**
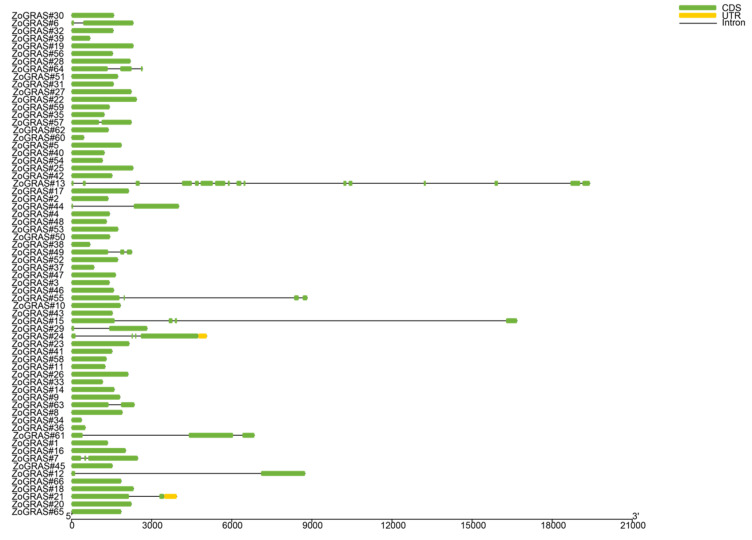
Gene structure of GRAS genes in ginger. Yellow boxes represent untranslated regions, green boxes represent exons, black lines represent introns.

**Figure 4 genes-14-00096-f004:**
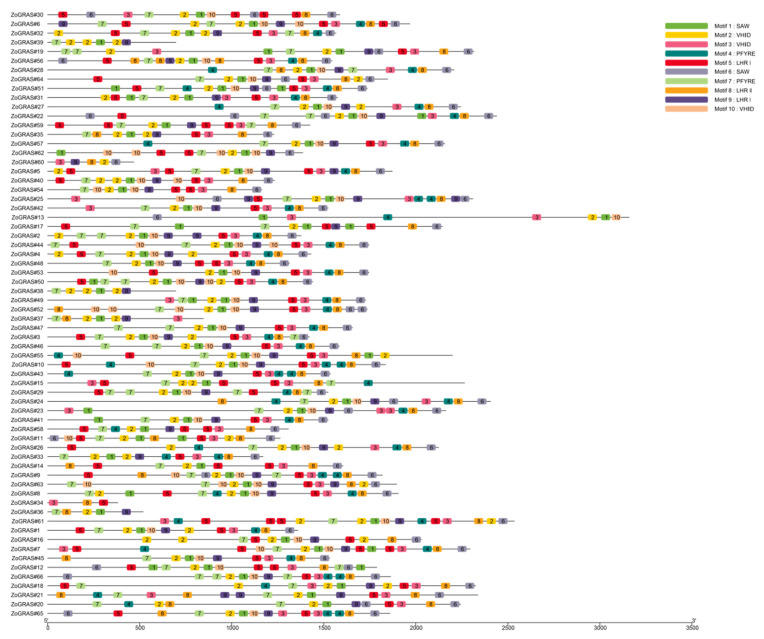
Motif composition of GRAS proteins in ginger. Colored boxes represent motifs detected in this study.

**Figure 5 genes-14-00096-f005:**
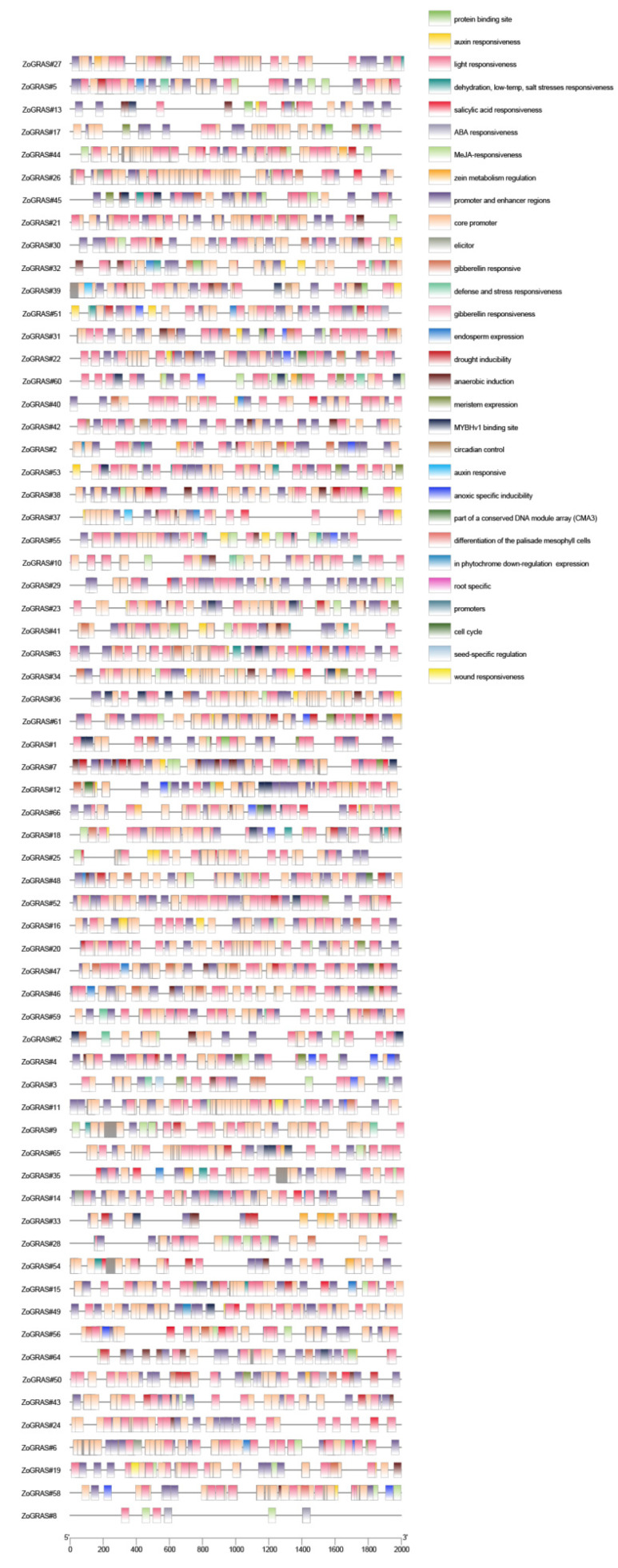
Cis-acting elements of ginger GRAS promoters. Different types of cis-acting elements are marked by different colors.

**Figure 6 genes-14-00096-f006:**
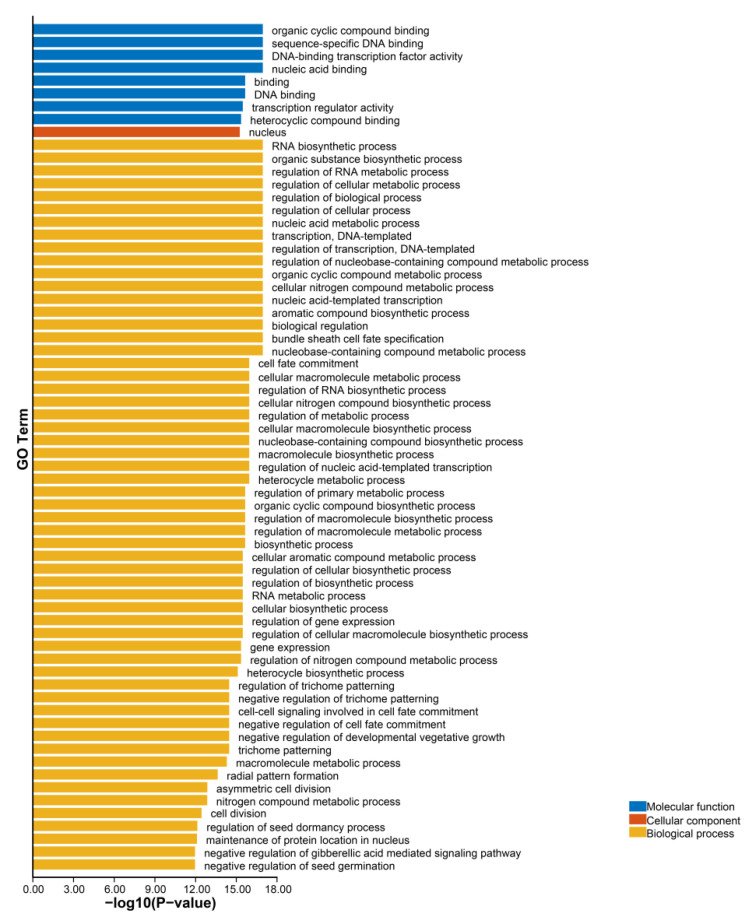
GO annotation of GRAS protein sequences.

**Figure 7 genes-14-00096-f007:**
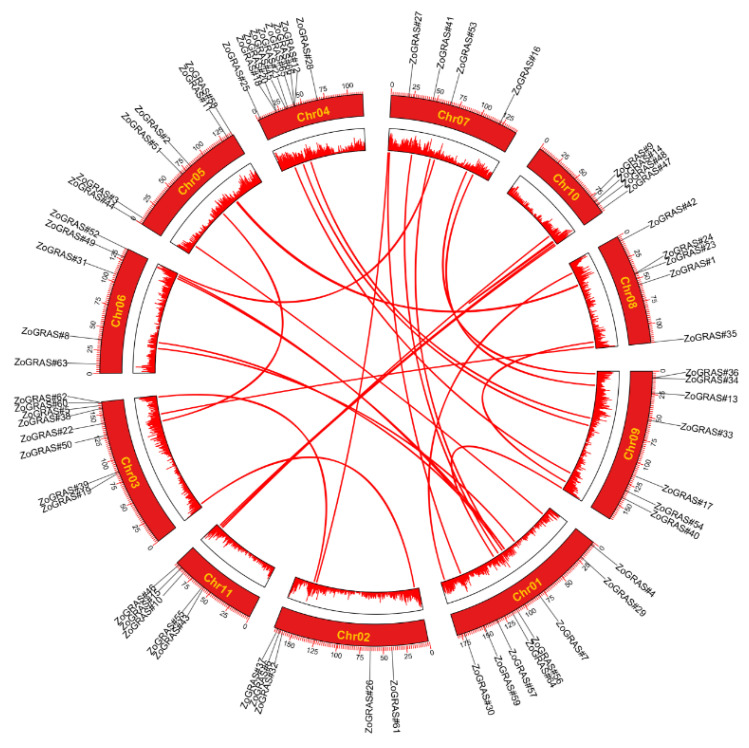
Diagram of chromosomal relationship of GRAS genes in ginger. The red lines represent duplicated GRAS gene pairs in ginger. The chromosome names of ginger are shown in the middle of each chromosome.

**Figure 8 genes-14-00096-f008:**
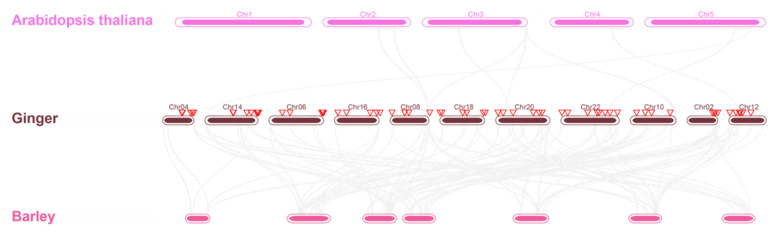
Collinearity analysis of ginger, Arabidopsis, and Barley. The red inverted triangles represents the *ZoGRAS* genes.

**Figure 9 genes-14-00096-f009:**
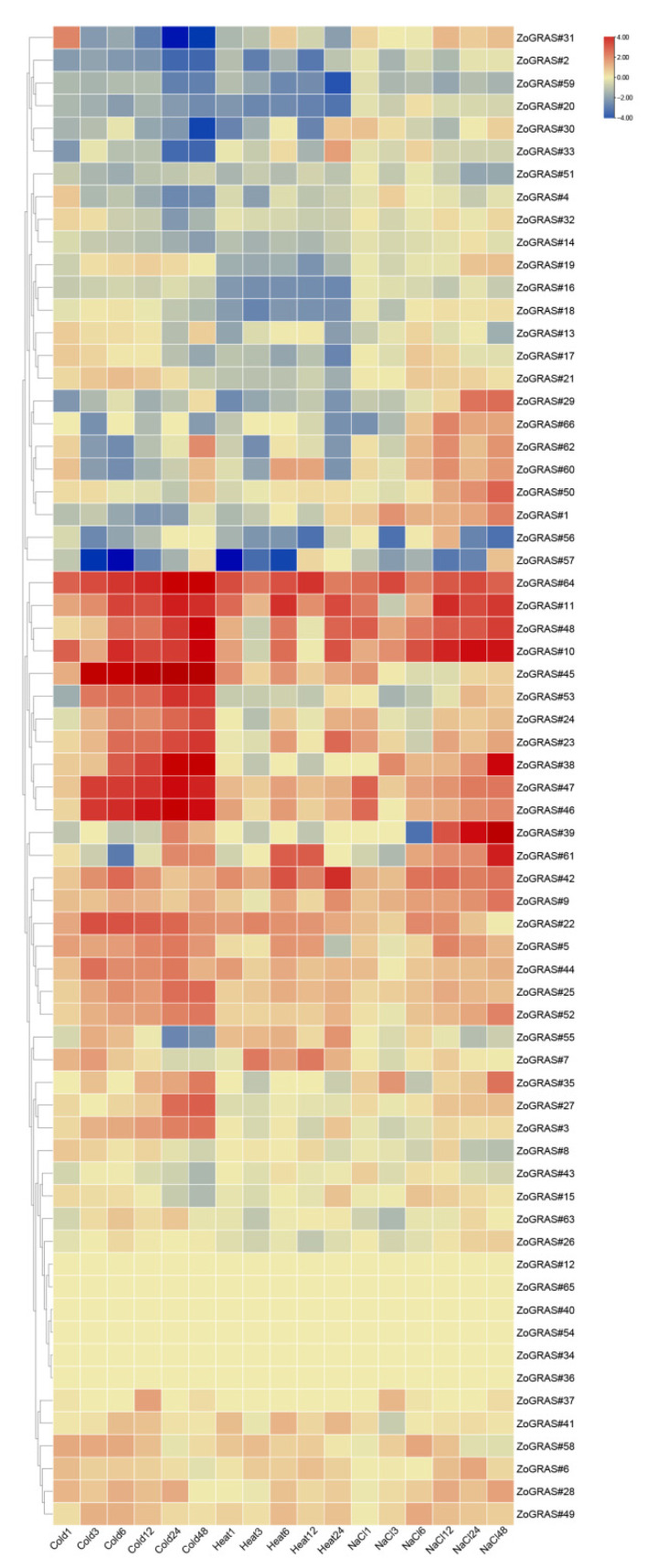
GRAS gene expression profile in ginger, including hierarchical clustering of GRAS gene expression profiles in ginger leaves under different stress conditions.

**Figure 10 genes-14-00096-f010:**
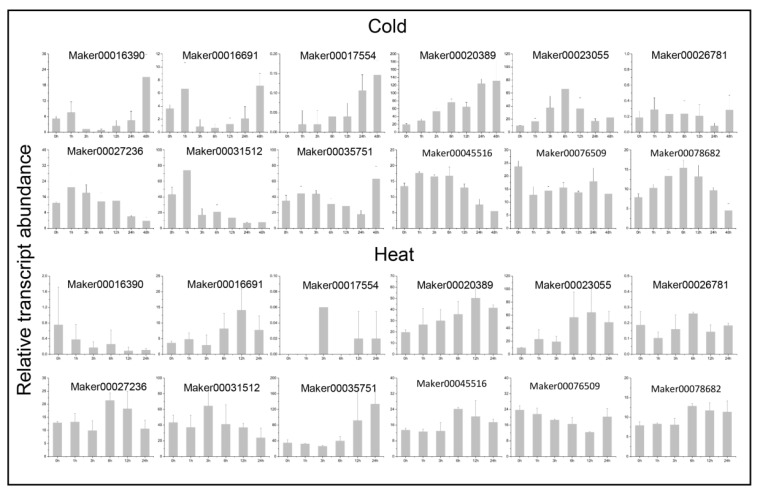
QRT-PCR expression analysis of GRAS genes under abiotic stresses. The data has been normalized, and the bar represents the standard deviation.

**Figure 11 genes-14-00096-f011:**
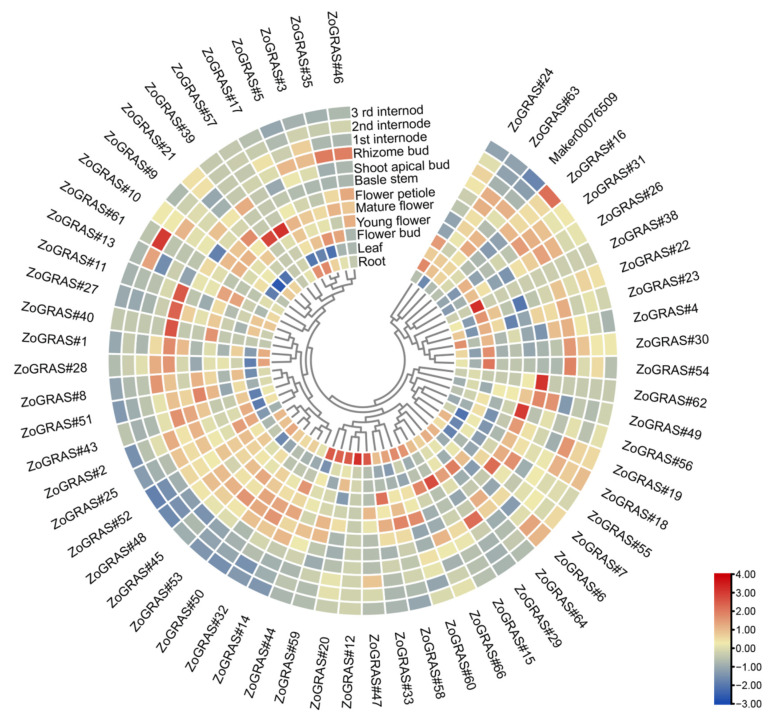
GRAS gene expression profile of ginger. Hierarchical clustering of GRAS gene expression profiles of ginger in 12 samples, including different tissues and developmental stages.

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
