# Peer review of "Genome-Wide Identification, Characterization, and Expression Analysis of GRAS Gene Family in Ginger (Zingiber officinale Roscoe)"

_genes, 2022, doi:10.3390/genes14010096_

Round 1

Reviewer 1 Report

This study provides genome wide identification and characterization of the GRAS Gene Family in Ginger which would be helpful for future studies and management of abiotic stress. However, there are some shortcomings which must be revised.

Line 13 replace stresses with stress.

Line 24 is very general it must be specifying. Mention specific results

Check line 32 “extraor dinary”

Line 33-34 lack citation. It's recommended to cite the following study.

https://doi.org/10.3390/genes13101699,

“than 3000 TFs have been demonstrated to be involved in various physiological processes and regulatory networks in plants” should be cited.

Add more details of economic and medicinal importance of the Ginger.

Until now which gene families have been identified and characterized in Ginger.

Section 2.4 is recommended to cite recent study.

https://doi.org/10.3390/ijms22179175,

Discussion line 381-383 any proof or previous study about this mechanism?

Include future recommendations in the conclusion.

Reviewer 2 Report

This manuscript titled “Genome-Wide Identification, Characterization, and Expression Analysis of GRAS Gene Family in Ginger (Zingiber officinale Roscoe)” presents essential new data about GRAS gene family. Materials and methods used in this study have been sufficiently described. It is worth emphasizing the author’s analyses importance of both cis‑acting elements in GRAS genes expression and gene expression analysis under different abiotic stress. However, adding miRNA-target analysis, WGCNA analysis, and experiments related to localization would be preferable for a standard manuscript.

Overall, this is an intriguing manuscript that presents the results of a well-planned and executed study. There are several shortcomings for that should be resolve.

Some minor remarks:

Line 20: “cis-acting”, “cis-regulatory” should be changed throughout the manuscript to “cis-acting”, “cis-regulatory”.

Line 152-157: What about qRT-PCR internal control? It is preferable to use two internal controls such as Ubiquitin, Actin, and so on. Please also include the primer list that you used in the experiments.

Line 159-160: A recent study that looked at cis-elements with a 2000bp promoter could be cited.

Line 196: “Chr04contained” should be “Chr04 contained”

Line 221 and Figure 3: Could you please attach an intron phase into the gene structure and describe it in the manuscript?

Line 124: It would be preferable if you kept the "Ginger-Arabidopsis" collinearity analysis in Figure 8 and skipped the "Ginger-Rice" collinearity analysis. Furthermore, adding more synteny pairs like "Ginger-Maize" and "Ginger-Barley" would be more convenient.

Line 354: “expression lev-els” should be “expression levels”

Suggestion 1: Could you please analyze the subcellular localization using webtools such as "Cell-PLoc 2.0," WoLF PSORT, PredSL, and so on? If possible, please include GFP localization to demonstrate that the prediction is valid.

Suggestion 2: Could you please analyze and include the GRAS miRNA target in the manuscript?

Suggestion 3: Could you please conduct a "Gene Set Enrichment Analysis" (GSEA) from the RNA-Seq data and include the results in the manuscript? It will help researchers better understand the biological role of the GRAS gene in ginger.
